# Diagnostic Role of ^18^F-PSMA-1007 PET/CT in Prostate Cancer Staging: A Systematic Review

**DOI:** 10.3390/diagnostics11030552

**Published:** 2021-03-19

**Authors:** Salam Awenat, Arnoldo Piccardo, Patricia Carvoeiras, Giovanni Signore, Luca Giovanella, John O. Prior, Giorgio Treglia

**Affiliations:** 1Institute for Radiology and Nuclear Medicine, University Hospital Knappschaftskrankenhaus Bochum, 44892 Bochum, Germany; ssawenat2001@yahoo.com; 2Department of Nuclear Medicine, E.O. Ospedali Galliera, 16128 Genoa, Italy; arnoldo.piccardo@galliera.it; 3School for Radiology Technicians, 6600 Locarno, Switzerland; patricia.carvoeiras@gmail.com; 4Department of Medicine, Università degli studi della Campania “L. Vanvitelli”, 81100 Caserta, Italy; giovanni.signore94@gmail.com; 5Clinic of Nuclear Medicine, Imaging Institute of Southern Switzerland, Ente Ospedaliero Cantonale, 6500 Bellinzona, Switzerland; luca.giovanella@eoc.ch; 6Department of Nuclear Medicine, University Hospital Zürich and University of Zürich, 8091 Zürich, Switzerland; 7Department of Nuclear Medicine and Molecular Imaging, Lausanne University Hospital, 1011 Lausanne, Switzerland; john.prior@chuv.ch; 8Faculty of Biology and Medicine, University of Lausanne, 1011 Lausanne, Switzerland; 9Academic Education, Research and Innovation Area, General Directorate, Ente Ospedaliero Cantonale, 6500 Bellinzona, Switzerland; 10Faculty of Biomedical Sciences, Università della Svizzera italiana, 6900 Lugano, Switzerland

**Keywords:** PET, PSMA, prostate cancer, staging, PSMA-1007, systematic review

## Abstract

Background: The use of prostate-specific membrane antigen (PSMA)-targeted agents for staging prostate cancer (PCa) patients using positron emission tomography/computed tomography (PET/CT) is increasing worldwide. We performed a systematic review on the role of ^18^F-PSMA-1007 PET/CT in PCa staging to provide evidence-based data in this setting. Methods: A comprehensive computer literature search of PubMed/MEDLINE and Cochrane Library databases for studies using ^18^F-PSMA-1007 PET/CT in PCa staging was performed until 31 December 2020. Eligible articles were selected and relevant information was extracted from the original articles by two authors independently. Results: Eight articles (369 patients) evaluating the role of ^18^F-PSMA-1007 PET/CT in PCa staging were selected. These studies were quite heterogeneous, but, overall, they demonstrated a good diagnostic accuracy of ^18^F-PSMA-1007 PET/CT in detecting PCa lesions at staging. Overall, higher primary PCa aggressiveness was associated with higher ^18^F-PSMA-1007 uptake. When compared with other radiological and scintigraphic imaging methods, ^18^F-PSMA-1007 PET/CT had superior sensitivity in detecting metastatic disease and the highest inter-reader agreement. ^18^F-PSMA-1007 PET/CT showed similar results in terms of diagnostic accuracy for PCa staging compared with PET/CT with other PSMA-targeted tracers. Dual imaging with multi-parametric magnetic resonance imaging and ^18^F-PSMA-1007 PET/CT may improve staging of primary PCa. Notably, ^18^F-PSMA-1007-PET/CT may detect metastatic disease in a significant number of patients with negative standard imaging. Conclusions: ^18^F-PSMA-1007 PET/CT demonstrated a good accuracy in PCa staging, with similar results compared with other PSMA-targeted radiopharmaceuticals. This method could substitute bone scintigraphy and conventional abdominal imaging for PCa staging. Prospective multicentric studies are needed to confirm these findings.

## 1. Introduction

Prostate cancer (PCa) is the second most common cancer among men, representing a common cause of cancer death worldwide [1].

A precise PCa staging through imaging methods is essential for correct disease management, as treatment options are different for localized PCa, locally advanced, or metastatic disease. At the same time, it should be taken into account in the selection process of the imaging techniques that these methods have progressively improved in the last years [2,3]. In the last decade, magnetic resonance imaging (MRI) has become the leading imaging modality in the primary detection and localization of PCa. Bone scintigraphy and conventional abdominal imaging are still recommended for staging intermediate- and high-risk PCa, but they are increasingly being replaced by new hybrid imaging modalities [2,3].

Advanced imaging methods including positron-emission tomography/computed tomography (PET/CT) and positron-emission tomography/magnetic resonance imaging (PET/MRI) using different radiopharmaceuticals have showed relevant results for early detection of local and systemic spread of PCa, according to available evidence-based data [4]. In particular, hybrid imaging methods combining functional and morphological information have the potential to overcome the limitations of conventional imaging methods in some clinical situations of PCa staging [2,3].

In this regard, PET/CT and PET/MRI using radiolabeled prostate specific membrane antigen (PSMA)-targeted agents are considered very useful imaging methods in detecting PCa lesions both at staging and at restaging [5,6,7,8,9,10].

PSMA is a transmembrane cell surface protein that is overexpressed in 95% of PCa cells; on the other hand, the overexpression of PSMA has not been found in benign prostatic diseases. However, PSMA is not prostate specific and is also found within other tissues than prostate and other tumors beyond PCa [11].

To date, several PSMA ligands slightly different in chemical structure are commercially available for diagnosis and therapy of PCa [11,12,13,14,15,16,17]. These molecules may be radiolabeled with different positron-emitters isotopes as Gallium-68 (^68^Ga), Fluorine-18 (^18^F), or Copper-64 (^64^Cu) to obtain PET radiopharmaceuticals, which can be used in the clinical practice or in clinical trials [11,12,13,14,15,16,17]. Currently, PSMA-targeted agents radiolabeled with ^68^Ga are the most used PSMA-based radiopharmaceuticals for PCa detection using PET imaging. More recently, PSMA ligands have been radiolabeled with other isotopes with more favorable physical characteristics compared with ^68^Ga, such as ^18^F or ^64^Cu [11,12,13,14,15,16,17].

Radiolabeling PSMA-targeted agents with ^18^F may provide several advantages compared with ^68^Ga radiolabeling for PET imaging, including a longer half-life of the radiopharmaceuticals and an improved resolution of PET images [11,12,13,14,15,16,17]. Among the different PSMA-targeted agents radiolabeled with ^18^F, ^18^F-PSMA-1007 has been recently introduced into the clinical practice [18] after successful preclinical studies [19].

A recent systematic review and meta-analysis demonstrated the good detection rate of PET/CT with PSMA-targeted agents radiolabeled with ^18^F (including ^18^F-PSMA-1007) in the setting of PCa restaging after primary treatment (as surgery or radiation treatment) [20].

Our aim is now to perform a systematic review of the literature to demonstrate the role of PET/CT with ^18^F-PSMA-1007 in PCa staging to add evidence-based data in this setting.

## 2. Materials and Methods

### 2.1. Search Strategy

Reporting of this systematic review conforms to the “Preferred Reporting Items for a Systematic Review and Meta-Analysis of Diagnostic Test Accuracy Studies” (PRISMA-DTA statement) and to specific guidelines for the systematic review of diagnostic accuracy studies [21,22].

A comprehensive computer literature search of PubMed/MEDLINE and Cochrane library databases was performed by two authors independently to find relevant published articles on the diagnostic role of ^18^F-PSMA-1007 PET/CT in PCa staging.

A search algorithm based on a combination of these terms was used: (A) “PSMA” AND (B) “1007” AND C) “prostate” OR “prostatic”. No beginning date limit nor language restrictions were used. The last update of the literature search was 31 December 2020. To expand the search, references of the retrieved articles were also screened, searching for additional studies.

### 2.2. Study Selection

Studies or subsets of studies investigating the diagnostic role of ^18^F-PSMA-1007 PET/CT in PCa staging were eligible for inclusion in the systematic review. The exclusion criteria were as follows: (a) articles outside the field of interest of this review (e.g., articles using ^18^F-PSMA-1007 PET/CT for PCa restaging after biochemical recurrence or for other indications than PCa staging or articles evaluating PCa staging with other PET tracers than ^18^F-PSMA-1007); (b) review articles, editorials, letters to editor, comments, and conference proceedings; and (c) case reports or small case series (less than five patients included).

Two authors independently reviewed the titles and abstracts of the retrieved articles using the abovementioned search algorithm. These authors have also applied the inclusion and exclusion criteria mentioned above and they have excluded articles if they were clearly ineligible. The same researchers then independently reviewed the full-text version of the remaining articles to assess their eligibility for inclusion in the systematic review resolving disagreements in a consensus virtual meeting.

### 2.3. Data Extraction

Two authors performed the data extraction from the eligible articles independently. Information was collected concerning basic study characteristics (authors, year of publication, country of origin, and study design), patient characteristics (type and number of PCa patients evaluated, mean/median age, Gleason score, and median prostate specific antigen (PSA) serum values before PET/CT), technical aspects (hybrid imaging modality used, fasting and hydration before radiotracer injection, mean radiotracer injected activity, time interval between radiotracer injection and image acquisition, PET/CT scan extension, PET/CT image analysis, and other imaging methods performed for comparison), and reference standard used. Data about change of management and diagnostic performance (sensitivity, specificity and accuracy) of ^18^F-PSMA-1007 PET/CT in PCa staging on a per patient- and on a per lesion-based analysis were also extracted.

### 2.4. Quality Assessment

The overall quality of the studies included in this systematic review was critically appraised based on the National Institute of Health (NIH) quality assessment tools (U.S. Department of Health & Human Services, Bethesda, MD, USA) [23].

## 3. Results

### 3.1. Literature Search

The literature search results are illustrated in Figure 1.

Performing a comprehensive computer literature search of PubMed/MEDLINE and Cochrane library database, 64 records were identified. These 64 records were screened and 56 articles were excluded after reviewing titles and abstracts: 33 because they were outside the field of interest of this review; 7 as reviews, editorials, or letters; 16 as case reports or small case series. Eight articles were selected and retrieved in full-text version. No additional studies were found after screening the references of these articles. Therefore, eight articles evaluating the diagnostic role of ^18^F-PSMA-1007 PET/CT in PCa staging were eligible for the systematic review [24,25,26,27,28,29,30,31]. The characteristics of these studies selected for the systematic review and including a total number of 369 PCa patients are presented in Table 1 and Table 2. The main findings of the articles included in the systematic review are summarized in Table 3 and described here below.

### 3.2. Qualitative Analysis (Systematic Review)

#### 3.2.1. Basic Study and Patient Characteristics

The included articles were published in the last years (range: from 2017 to 2020). Different countries from Europe, Asia, South America, and Africa were represented. Concerning the study design, most of the articles were retrospective (5 out of 8) and single centre (6 out of 8) studies. Patients with both intermediate- and high-risk PCa were included in the selected studies. The median age of the PCa patients included in these studies ranged from 65 to 69.5 years. The Gleason score (GS) of PCa patients largely varied among the included studies: GS was 6 in 0–12%, 7 in 30–69%, 8 in 0–3%, and >8 in 0–70% of cases. Median PSA serum values before PET/CT among the included PCa patients ranged from 6 to 85 ng/mL.

#### 3.2.2. Technical Aspects

The hybrid imaging modality was PET/CT without contrast-enhanced CT in most of the cases (7 out of 8 articles). Notably, there were no included articles evaluating ^18^F-PSMA-1007 PET/MRI in PCa staging. Fasting and hydration before radiotracer injection has been reported in a few articles. The mean injected radiotracer activity ranged from 240 to 291 MBq (4 MBq/kg) in the included studies. The time interval between radiotracer injection and PET/CT image acquisition was quite different among the included studies, ranging from 60 to 180 min. In some studies, a dual time point PET/CT imaging was carried out. PET/CT scan extension also varied among the studies. The PET image analysis was performed by using qualitative (visual) analysis and additional semi-quantitative analysis through the calculation of the maximal standardized uptake values (SUV_max_) in all the studies. At visual analysis, all foci of radiotracer uptake greater than the surrounding tissue that could not be explained by physiological activity were considered to be abnormal.

#### 3.2.3. Quality Assessment

The study quality was judged as fair for all the included articles according to the NIH quality assessment tool (Table 3) [23].

#### 3.2.4. Main Findings

No adverse events due to ^18^F-PSMA-1007 administration were reported in the included studies [24,25,26,27,28,29,30,31]. Concerning the radiation dosimetry, with an effective dose of 4.4–5.5 mSv for an injected activity of 200–250 MBq, ^18^F-PSMA-1007 is similar to other PSMA-targeted PET agents as well as to other ^18^F-labelled PET radiopharmaceuticals.

In comparison with other PSMA-targeted PET radiopharmaceuticals, ^18^F-PSMA-1007 has similar biodistribution (with physiological uptake in liver, spleen, salivary glands, lacrimal glands, small intestine, pancreas, and kidneys), but reduced urinary clearance (enabling excellent assessment of the prostate) and increased hepatobiliary clearance [25,29]. Favorable tumor-to-background ratios are usually observed 2–3 h after ^18^F-PSMA-1007 injection [25].

A significant positive correlation between PSA level/GS and ^18^F-PSMA-1007 uptake in primary PCa tumors has been demonstrated [27]. Overall, higher tumor aggressiveness was associated with higher ^18^F-PSMA-1007 uptake [27,30]. SUV_max_ of prostatic tumor in high-risk PCa patients was significantly higher than those in intermediate-risk PCa patients [27].

The included studies using a different reference standard demonstrated a good diagnostic accuracy of ^18^F-PSMA-1007 PET/CT in detecting PCa lesions at staging both on a per patient- and on a per lesion-based analysis (Table 3). The most frequent sites of PCa lesions detected by ^18^F-PSMA PET/CT in PCa staging were the primary tumor, regional and distant lymph nodal metastases, and bone metastases [24,25,26,27,28,29,30,31].

Overall, the sensitivity of ^18^F-PSMA-1007 PET/CT ranged from 74% to 100% on a per patient-based analysis and from 71% to 100% on a per lesion-based analysis; the specificity of ^18^F-PSMA-1007 PET/CT ranged from 76% to 100% on a per patient-based analysis and from 91% to 100% on a per lesion-based analysis; the accuracy of ^18^F-PSMA-1007 PET/CT ranged from 80% to 100% on a per patient-based analysis and from 93% to 95% on a per lesion-based analysis [24,25,26,27,28,29,30,31].

When compared with other radiological and scintigraphic imaging methods, ^18^F-PSMA-1007 PET/CT had superior sensitivity in detecting metastatic diseases and the highest inter-reader agreement [24].

When compared with PET/CT with other PSMA-targeted tracers (^18^F-DCFPyL and ^68^Ga-PSMA), ^18^F-PSMA-1007 PET/CT showed similar results in terms of diagnostic accuracy for PCa staging with an almost perfect concordance in PCa lesion detection among the different PSMA-targeted tracers, but with an overall higher uptake in PCa lesions (measured as SUV_max_) compared with other PSMA-targeted agents [26,29].

Overall, ^18^F-PSMA-1007 PET/CT, multi-parametric MRI, and final histopathology correlated well in PCa tumors [28,30]. ^18^F-PSMA-1007 PET/CT provides accurate primary PCa staging (T-staging). In this regard, dual imaging with multi-parametric MRI and ^18^F-PSMA-1007 PET/CT may improve staging of primary PCa [28,30].

For lymph nodal staging (N-staging), it has been demonstrated that ^18^F-PSMA-1007 PET/CT is able to detect lymph nodal metastases of PCa, even those small in size, with high specificity [31].

For detection of distant metastases of PCa (M-staging), the accuracy of ^18^F-PSMA-1007 PET/CT for detection of bone metastases was higher than bone scintigraphy (with planar and tomographic acquisitions), CT, and whole-body MRI [24]. Notably, ^18^F-PSMA-1007 PET/CT may detect metastatic disease in a significant number of patients with negative standard imaging for metastatic spread of PCa, changing the clinical management in about 20% of PCa patients at staging [24].

## 4. Discussion

### 4.1. Discussion of Main Findings

^18^F-PSMA-1007 is a novel PET radiopharmaceutical recently introduced into clinical practice to evaluate patients with PCa [25]. Some studies have evaluated the diagnostic role of ^18^F-PSMA-1007 PET/CT for PCa staging and restaging. To the best of our knowledge, currently, there are no published systematic reviews on the role of ^18^F-PSMA-1007 PET/CT for PCa staging, whereas a systematic review on the role of ^18^F-PSMA-targeted agents in PCa restaging is already available [20]. Therefore, we have collected and summarized published data through a systematic review approach [22,32] to add more evidence-based data on the role of ^18^F-PSMA-1007 PET/CT for PCa staging.

Compared with other PSMA-targeted PET radiopharmaceuticals, the effective dose of ^18^F-PSMA-1007 PET/CT was comparable (4.4–5.5 mSv for 200–250 MBq) [25]. This is not surprising as the radiation exposure for this PET radiopharmaceutical depends primarily on the short physical half-life of the isotope (^18^F) rather than the biological half-life of the carrier molecule [25].

^18^F-PSMA-1007 showed a similar biodistribution to normal organs compared with other PSMA-targeted tracers labeled with ^18^F or ^68^Ga, and this is not surprising because of the same target molecule of these radiopharmaceuticals (PSMA) [25]. More importantly, the increased uptake of this radiopharmaceutical in PCa lesions compared with other PSMA-targeted tracers improves tumor-to-background ratios, making it easier to detect small PCa lesions [25,26,29]. Another clear advantage of ^18^F-PSMA-1007 compared with other PSMA-targeted PET tracers is its reduced urinary clearance (^18^F-PSMA-1007 has increased hepatobiliary clearance and it is temporarily retained in the kidney parenchyma). These characteristics allow an excellent assessment of the prostate and pelvic region owing to the reduced interference of the urinary radioactivity and, therefore, an excellent local staging of PCa. This benefit accrues from decreased tracer in ureters (which can confound nodal assessment) as well as bladder (improved visualization of prostate) [25,26,29]. Owing to its hepatobiliary excretion, a higher radiopharmaceutical uptake in the liver is observed with ^18^F-PSMA-1007 compared with other PSMA-targeted PET tracers; therefore, liver metastases of PCa may be missed by ^18^F-PSMA-1007 PET/CT, but this may happen in late PCa disease stages when rare cases of liver metastases can occur [25,26,29]. Interestingly, a dietary preparation of PCa patients by fasting or the attempt to clear liver activity by high caloric drinks does not have a significant effect on ^18^F-PSMA-1007 uptake in the liver or in the small bowel [33].

^18^F-PSMA-1007 can be produced in large amounts in PET centers with an onsite cyclotron; the longer half-life of ^18^F (110 min) compared with ^68^Ga (68 min) enhances the transfer to satellite centers for the radiopharmaceutical administration [25]. Conversely, radiolabeling PSMA with ^68^Ga can be performed in imaging centers using a ^68^Ge/^68^Ga radionuclide generator without its own cyclotron. However, a single preparation of ^68^Ga may only allow scanning in a limited number of patients and the output depends on the half-life of the generator [29]. Radiolabeling PSMA-targeted agents with ^18^F should also allow to obtain a spatial resolution better than that of ^68^Ga, on the basis of the nuclear physical properties of these radioisotopes [34,35].

Because of the longer half-life of ^18^F compared with ^68^Ga, delayed imaging is possible using the radiofluorinated compounds. In particular, ^18^F-PSMA-1007 demonstrated a remarkable increase in radiopharmaceutical uptake (measured as SUV_max_) in PCa lesions when PET imaging was postponed until 3 h after radiopharmaceutical injection [25,26,28]. The studies included in this systematic review largely varied about the time between radiopharmaceutical injection and PET/CT image acquisition, which ranged between 60 and 180 min. We could hypothesize that 120 min after radiopharmaceutical injection as acquisition time for ^18^F-PSMA-1007 PET could be a good balance between optimal PCa lesion contrast and optimal patient throughput in clinical practice, but this hypothesis needs to be verified through high-quality studies [26].

A positive correlation between the intensity of ^18^F-PSMA-1007 uptake and the GS/PSA level in the primary tumors of PCa patients was found [27]. Furthermore, the SUV_max_ of the primary tumor was valuable for identifying high-risk PCa [27]. A timely and accurate diagnosis of high-risk PCa is relevant for the clinicians owing to the related prognostic implications and the different clinical management among the various PCa risk categories [36]. The PCa risk classifications are based on clinical stage, GS by biopsy, and PSA level before treatment [36]. However, the assessment of GS in patients who have undergone random transrectal ultrasound-guided (TRUS) biopsy may be not reliable; furthermore, some patients may refuse biopsy for a variety of reasons, and assessing disease by digital rectal examination has significant inter-observer variability. PET/CT with PSMA-targeted agents, as a noninvasive imaging method, may compensate for these shortcomings and may be helpful for clinicians to reliably diagnose high-risk PCa, especially when the patients are unable to receive biopsy or the histology results of biopsy are not satisfactory [27].

Overall, the studies included in our systematic review demonstrated a good diagnostic accuracy of ^18^F-PSMA-1007 PET/CT in detecting PCa lesions at staging both on a per patient- and on a per lesion-based analysis, even if a different reference standard was used in the included studies (Table 3) [24,25,26,27,28,29,30,31]. In particular, when compared with other radiological and scintigraphic imaging methods, ^18^F-PSMA-1007 PET/CT had superior sensitivity in detecting metastatic diseases and the highest inter-reader agreement [24]. Staging of PCa is essential to determine the most appropriate treatment modality and exclusion of metastases in PCa is paramount as local radical therapy cannot achieve cure in the presence of distant metastases and may instead expose patients to treatment-related adverse effects without any therapeutic benefit [36]. ^18^F-PSMA-1007 PET/CT increases the detection of metastatic disease compared with standard conventional imaging, influencing the clinical management in about 20% of PCa patients at staging [24]. We expect that, in the near future, bone scintigraphy and conventional abdominal imaging will be gradually replaced by PSMA-based hybrid imaging for a more accurate staging of intermediate- and high-risk PCa. In this regard, the recently published proPSMA study robustly supports the use of PSMA-based hybrid imaging instead of conventional bone scan and abdominal CT for the initial staging of PCa [37].

Nevertheless, false negative and false positive findings of ^18^F-PSMA-1007 PET/CT in detecting PCa lesions should be taken into account. Previous cancer therapies, low PSMA expression caused by tumor heterogeneity, or PCa lesions containing only small amounts of cancer cells might be responsible for false-negative ^18^F-PSMA PET/CT results in some PCa patients; false negative findings may be also due to PCa lesions located in sites of physiological radiopharmaceutical uptake [24,25,26,27,28,29,30,31]. False positive findings may be due to benign lesions or other malignancies than PCa with PSMA overexpression [24,25,26,27,28,29,30,31]. Interestingly, one of the most reported false positive findings of ^18^F-PSMA-1007 PET/CT was related to moderately increased bone uptake without any corresponding anatomical findings in primary or follow-up imaging [24].

To date, multi-parametric MRI in combination MRI/TRUS fusion biopsy outperforms the standard TRUS-guided biopsy and plays a pivotal role for PCa detection and local staging [36]. Compared with multi-parametric MRI, ^18^F-PSMA-1007 PET/CT showed similar accuracy and general concordance for local detection and staging of PCa, but the higher resolution and anatomic landmark definition derived from mpMRI should be underlined [28,30]. Preliminary data showed that ^18^F-PSMA-1007 PET/CT observed seminal vesicle invasion of PCa better than multi-parametric MRI; in contrast, extracapsular extension of PCa was better detected by multi-parametric MRI compared with ^18^F-PSMA-1007 PET/CT [30]. Interestingly, the most promising results were achieved by combining ^18^F-PSMA-1007 PET/CT and multi-parametric MRI, suggesting that a combination of these methods would improve both PCa detection and the local staging [28,30]. In this regard, ^18^F-PSMA-1007 PET/MRI could be the future one-stop-shop imaging modality for PCa staging, even if, to date, the availability of this method is still limited [28,30].

The results of preliminary studies comparing ^18^F-PSMA-1007 PET/CT with other PSMA-targeted tracers labeled with ^18^F or ^68^Ga showed good concordance between these radiotracers; all PSMA-targeted radiotracers performed equally well and showed high accuracy in detection of PCa lesions [26,29]. In view of their near-equal performance, these preliminary studies suggest the routine-use of ^18^F-PSMA-1007 in lieu of other PSMA-based PET tracers for staging PCa patients, in particular for the reduced urinary clearance and the better target-to-background uptake ratio of ^18^F-PSMA-1007. Nevertheless, PSMA-based PET radiotracers can be selected mainly based on the local availability [26,29].

### 4.2. Limitations

This systematic review has several limitations. First of all, a limited number of articles on the diagnostic role of ^18^F-PSMA-1007 PET/CT in PCa staging were available and some of them were retrospective studies (with possible selection bias) and/or included a low number of PCa patients.

Second, the included studies were quite heterogeneous for several aspects including basic studies and patient characteristics (Table 1), technical aspects of PET/CT (Table 2), or main outcome measures (Table 3). Owing to this proven heterogeneity, we did not perform a meta-analysis on the diagnostic accuracy of ^18^F-PSMA-1007 PET/CT in PCa staging, taking into account that the heterogeneity among studies would be a relevant source of bias [21,22].

The lack of histological verification of suspected distant metastatic lesions detected by ^18^F-PSMA-1007 PET/CT is an inherent limitation for some studies included in this systematic review. In the absence of histological validation, it cannot be excluded that some lesions detected with ^18^F-PSMA-1007 PET/CT may represent false-positive findings. Nevertheless, if modern imaging methods are performed in PCa patients, then confirmation of positive findings is needed only in selected cases and with a biopsy when findings are equivocal [3].

According to published data, ^18^F-PSMA-1007 PET/CT increases the detection of metastatic disease in PCa, changing the management in a significant number of cases. Nevertheless, the ultimate benefit of earlier metastasis detection upon prognosis and survival of PCa patients has yet to be established.

### 4.3. Suggestions for Future Studies

Overall, our systematic review underlined the increasing diagnostic role of ^18^F-PSMA-1007 PET/CT in PCa staging, but large prospective multicentre studies are needed to strengthen the role of this imaging method in this setting. In particular, more studies comparing ^18^F-PSMA-1007 PET/CT with other imaging methods used in PCa staging are warranted. Furthermore, more studies on the comparison between ^18^F-PSMA-1007 and other PSMA-targeted PET radiopharmaceuticals are expected in the future. To date, there are no published studies on ^18^F-PSMA-1007 PET/MRI in PCa staging. As this hybrid imaging method could be an effective one-stop-shop imaging modality for PCa staging, future research should be focused on this topic.

## 5. Conclusions

^18^F-PSMA-1007 PET/CT demonstrated good accuracy in PCa staging, with similar results compared with other PSMA-targeted PET radiopharmaceuticals. This imaging method could substitute bone scintigraphy and conventional abdominal imaging for staging of intermediate- and high-risk PCa. Owing to the limited data available so far, further studies and particularly prospective multicentre studies are needed to strengthen the diagnostic role of ^18^F-PSMA-1007 PET/CT in PCa staging.

## Figures and Tables

**Figure 1 diagnostics-11-00552-f001:**
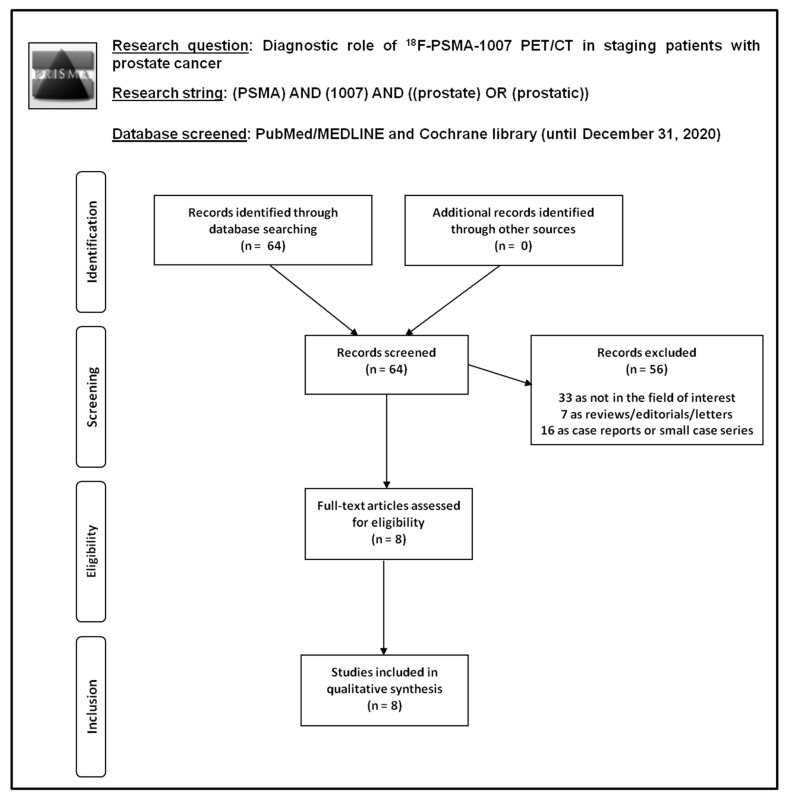
Flow chart of the search for eligible studies on the diagnostic role of ^18^F-PSMA-1007 positron emission tomography/computed tomography (PET/CT) in prostate cancer staging. PSMA, prostate-specific membrane antigen.

**Table 1 diagnostics-11-00552-t001:** Basic study and patient characteristics.

Authors	Year	Country	Study Design	Type Of Patients Evaluated	PCa Patients Performing ^18^F-PSMA-1007 PET/CT	Mean/Median Age (Years)	Gleason Score (Percentage)	Median PSA Values Before PET/CT (ng/ml) (Range)
Anttinen et al. [24]	2020	Finland	Prospective single-center	Patients with high-risk PCa at staging	80	Mean: 70 ± 7	GS 6: 3 (4%) GS 7: 30 (37%) GS 8: 13 (16%) GS 9–10: 34 (43%)	12 (3–2000)
Giesel et al. [25]	2017	Germany	Retrospective single-center	Patients with high-risk PCa at staging	10	Median: 65 (55–77)	GS 7: 3 (30%) GS 8: 2 (20%) GS 9: 5 (50%)	14 (5.8–87.3)
Giesel et al. [26]	2018	Germany and South Africa	Prospective bicentric	Patients with PCa at staging	12	Median: 66 (54–82)	GS 6: 1 (8%) GS 7: 5 (42%) GS 8: 4 (33%) GS 9: 2 (17%)	85 (10–279.8)
Hong et al. [27]	2020	China	Retrospective single-center	Patients with non-metastatic intermediate- or high-risk PCa at staging	101	Median: 69 (43–87)	GS 6: 4 (4%) GS 7: 67 (66%) GS 8: 9 (9%) GS 9: 21 (21%)	11.1 (0.97–178.2)
Kesch et al. [28]	2017	Germany	Retrospective single-center	Patients with high-risk PCa at local staging	10	Median: 67 (55–77)	GS 7: 3 (30%) GS 9: 7 (70%)	13.1 (5.8–40)
Kuten et al. [29]	2020	Israel	Prospective single-center	Patients with intermediate- or high-risk PCa at staging	16	Median: 68 (56–74)	GS 6: 2 (12%) GS 7: 11 (69%) GS 8: 3 (19%)	6.35 (3.5–14.44)
Privé et al. [30]	2020	Netherlands	Retrospective single-center	Patients with intermediate- or high-risk PCa at staging	53	Median: 67	GS 6: 5 (9%) GS 7: 19 (36%) GS 8: 12 (23%) GS 9–10: 17 (32%)	12
Sprute et al. [31]	2020	Germany, Chile, Japan	Retrospective multicenter	Patients with PCa at staging	87	Median: 69 (48–78)	GS 6: 3% GS 7: 60% GS 8: 7% GS 9: 30%	11.7 (0.1–120)

Legend: CT = computed tomography; GS = Gleason score; PCa = prostate cancer; PET = positron emission tomography; PSA = prostate specific antigen; PSMA = prostate specific membrane antigen.

**Table 2 diagnostics-11-00552-t002:** Technical aspects of ^18^F-PSMA-1007 PET/CT in the included studies.

Authors	Hybrid Imaging Modality	Fasting/Hydration before Radiotracer Injection	Mean Radiotracer Injected Activity (MBq) (Range)	Time Interval between Radiotracer Injection and Image Acquisition (Minutes)	PET/CT Scan Extension	Image Analysis	Other Imaging Performed for Comparison
Anttinen et al. [24]	PET/CT with low-dose CT	NR/NR	263 ± 27 (205–355)	60	From vertex to mid-thigh	visual and semi-quantitative (SUV_max_)	Planar scintigraphy and SPECT/CT with ^99m^Tc-diphosphonates, CT, WBMRI with DWI
Giesel et al. [25]	PET/CT with low-dose CT	NR/NR	275 (111–356)	60 + 180	NR	visual and semi-quantitative (SUV_max_)	–
Giesel et al. [26]	PET/CT with low-dose CT	4 h/NR	240–260	120	From vertex to thighs	visual and semi-quantitative (SUV_max_)	^18^F-DCFPyL PET/CT
Hong et al. [27]	PET/CT with low-dose CT	NR/NR	4 MBq/kg; 291 (185–366)	120	From skull base to mid-thigh	visual and semi-quantitative (SUV_max_)	–
Kesch et al. [28]	PET/CT with low-dose CT	NR/NR	NR	60 + 180	NR	visual and semi-quantitative (SUV_max_)	multi-parametric MRI
Kuten et al. [29]	PET/CT with low-dose CT	NR/yes	4 MBq/kg	60	From vertex to mid-thigh	visual and semi-quantitative (SUV_max_)	^68^Ga-PSMA-11 PET/CT
Privé et al. [30]	PET/CT with contrast enhanced CT	NR/yes	~250	90 ± 10	NR	visual and semi-quantitative (SUV_max_)	multi-parametric MRI
Sprute et al. [31]	PET/CT with low-dose CT	NR/NR	270	90 (47–169)	NR	visual and semi-quantitative (SUV_max_)	–

Legend: CT = computed tomography; DWI = diffusion weighted imaging; MBq = MegaBecquerel; MRI = magnetic resonance imaging; NR = not reported; PET/CT = positron emission tomography/computed tomography; SPECT = single photon emission computed tomography; SUV_max_ = maximal standardized uptake value; ^99m^Tc = technetium-99m; WBMRI = whole-body magnetic resonance imaging.

**Table 3 diagnostics-11-00552-t003:** Main findings of the included studies about ^18^F-PSMA-1007 PET/CT in staging patients with prostate cancer.

Authors	Reference Standard	Patient-Based Analysis			Lesion-Based Analysis			Change of Management by Using PET/CT	Study Quality *	Study Summary
		Sensitivity	Specificity	Accuracy	Sensitivity	Specificity	Accuracy			
Anttinen et al. [24] ***	The reference standard diagnosis was defined using all available information accrued during at least 12 months of clinical follow-up	86–95%	76–90%	80–89%	72–86%	NR	NR	14/79 (18%)	fair	^18^F-PSMA-1007 PET/CT has superior sensitivity and the highest inter-reader agreement compared with standard and advanced imaging modalities for PCa staging.
Giesel et al. [25]	NR	100%	100%	100%	95%	100%	NR	NR	fair	^18^F-PSMA-1007 PET/CT has high sensitivity in detecting PCa lesions.
Giesel et al. [26]	NR	100%	NR	NR	NR	NR	NR	NR	fair	Excellent imaging quality and concordance are achieved with both ^18^F-DCFPyL and ^18^F-PSMA-1007.
Hong et al. [27] ***	Histology	NR	NR	NR	NR	NR	NR	NR	fair	There is a significant positive correlation between PSA level/GS and SUV_max_ at ^18^F-PSMA-1007 PET/CT.
Kesch et al. [28]	Histology and mpMRI	100%	NR	NR	93%	92%	93%	NR	fair	Comparison with histopathology demonstrates that ^18^F-PSMA-1007 PET/CT is promising for accurate local staging of PCa.
Kuten et al. [29]	Histology	100%	NR	NR	100%	91%	95%	NR	fair	Both ^18^F-PSMA-1007 and ^68^Ga-PSMA-11 PET/CT may identify all dominant prostatic lesions in patients with PCa at staging. ^18^F-PSMA-1007 may detect additional lesions of limited clinical relevance.
Privé et al. [30] ***	Histology and mpMRI	98%	NR	NR	NR	NR	NR	NR	fair	Dual imaging with mpMRI and ^18^F-PSMA-1007 PET/CT may improve staging of primary PCa. Higher PCa aggressiveness was associated with higher SUV_max_ at ^18^F-PSMA-1007 PET/CT.
Sprute et al. [31] ***	Histology	74% **	99% **	NR	71% **	99.5% **	NR	NR	fair	^18^F-PSMA-1007 PET/CT reliably detects PCa lymph nodal metastases with very high specificity.

Legend: * = according to the National Institute of Health (NIH) quality assessment tools; ** = lymph nodal lesions only; *** = studies with larger patient population; GS = Gleason score; mpMRI = multi-parametric magnetic resonance imaging; NR = not reported; PCa = prostate cancer; PET/CT = positron emission tomography/computed tomography; PSA = prostate specific antigen; PSMA = prostate specific membrane antigen; SUV_max_ = standardized uptake value.

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
