# Peer review of "Diagnostic Role of 18F-PSMA-1007 PET/CT in Prostate Cancer Staging: A Systematic Review"

_diagnostics, 2021, doi:10.3390/diagnostics11030552_

Round 1

Reviewer 1 Report

This is a succinct review of PSMA 1007 for the primary staging of prostate cancer.   As the number of studies utilizing PSMA 1007 is still relatively small, it would have been nice to see both the primary and recurrent studies summarized.  The authors reference an existing metanalysis of 18F PSMA for restaging but this study wasn't exclusively PSMA 1007 based.

I feel the paper would be enhanced through some minor additions.  

Firstly in the tables there should be some indication of what "gold standard" was used within each study (i.e. histology vs. clinical surrogate etc).

Secondly, the authors appropriately flag the lower renal clearance of the tracer as an advantage;  this benefit accrues from decreased tracer in ureters (which can confound nodal assessment) as well as bladder (improved visualization of prostate).

Finally, in table 3 where performance is compared it would be helpful to include the number of patients in each study to remind the reader of the larger studies when looking at the results

Overall a very nice review of a promising new PSMA tracer!

Author Response

We have revised the manuscript taking into account the comments of the reviewer:

1) In table 3 we have added the reference standard used/reported in each included study.

2) We have added the statement "this benefit accrues from decreased tracer in ureters (which can confound nodal assessment) as well as bladder (improved visualization of prostate)" in the discussion of the revised manuscript.

3) In table 3 we have indicated the studies with a larger patient population.

Reviewer 2 Report

The authors performed a systematic review of the available studies that used PSMA-1007 for the initial staging of prostate cancer. Eight studies involving 369 patients were selected and analyzed.

The study is very well written, the topic is of clinical interest, and I think that the readers of Diagnostics will highly appreciate this content. The methodology is clearly presented, and the interpretation of selected articles is correct.

No major issues from my side.

I just propose a few suggestions about parts of the manuscript whose revision might potentially improve the reported data's understanding and readability.

 1) Even if the review focuses on PSMA-1007 it is mandatory to introduce and briefly discuss the proPSMA study (PMID: 32209449), as it robustly supports the use of PSMA imaging instead of conventional bone scan and abdominal CT for the initial staging of prostate cancer.

2) The reference standard for calculating PSMA-1007 diagnostic accuracy should be indicated in table 3 and more extensively commented on in the text (when available).

3) Discussion, line 314: it is not abundantly clear on what bases the authors suggest using 120 minutes as the ideal time interval between tracer injection and image acquisition.    

Author Response

We have revised the manuscript taking into account the reviewer' suggestions.

 1) We have added in the discussion of the revised manuscript that the proPSMA study robustly supports the use of PSMA imaging instead of conventional bone scan and abdominal CT for the initial staging of prostate cancer. We have also added this study in the reference list.

2) We have added the reference standard used for calculating PSMA-1007 PET/CT diagnostic accuracy in table 3, citing the results in the text.

3) We have modified the statement about the acquisition at 120 min after radiopharmaceutical injection. We have now reported in the revised manuscript that "We could hypothesize that 120 minutes after radiopharmaceutical injection as acquisition time for 18F-PSMA-1007 PET could be a good balance between optimal PCa lesion contrast and optimal patient throughput in clinical practice, but this hypothesis needs to be verified through high-quality studies".